# Increasing Protein Content of Rice Flour with Maintained Processability by Using Granular Starch Hydrolyzing Enzyme

**DOI:** 10.3390/molecules28083522

**Published:** 2023-04-17

**Authors:** Jinxing Zhai, Xiaoxiao Li, Birte Svensson, Zhengyu Jin, Yuxiang Bai

**Affiliations:** 1State Key Laboratory of Food Science and Technology, Jiangnan University, Wuxi 214122, China; 2Enzyme and Protein Chemistry, Department of Biotechnology and Biomedicine, Technical University of Denmark, DK-2800 Kongens Lyngby, Denmark; 3School of Food Science and Technology, Jiangnan University, Wuxi 214122, China; 4Collaborative Innovation Center of Food Safety and Quality Control in Jiangsu Province, Jiangnan University, Wuxi 214122, China; 5International Joint Laboratory on Food Safety, Jiangnan University, Wuxi 214122, China

**Keywords:** rice flour, rice protein, granular starch hydrolyzing enzyme, hydrolytic mechanism, physicochemical property

## Abstract

Rice flour (RF) has become a promising food material. In the present study, RF with higher protein content was prepared using a granular starch hydrolyzing enzyme (GSHE). Particle size, morphology, crystallinity, and molecular structures of RF and rice starch (RS) were characterized to establish a hydrolytic mechanism; thermal, pasting, and rheological properties were determined to evaluate processability using differential scanning calorimetry (DSC), rapid viscosity analysis (RVA), and rheometer, respectively. The GSHE treatment resulted in pinholes, pits, and surface erosion through sequential hydrolysis of crystalline and amorphous areas on the starch granule surface. The amylose content decreased with hydrolysis time, while the very short chains (DP < 6) increased rapidly at 3 h but decreased slightly later. After hydrolysis for 24 h, the protein content in RF increased from 8.52% to 13.17%. However, the processability of RF was properly maintained. Specifically, the data from DSC showed that the conclusion temperature and endothermic enthalpy of RS barely changed. The result of rapid RVA and rheological measurement indicated that RF paste viscosity and viscoelastic properties dropped rapidly after 1 h hydrolysis and thereafter recovered slightly. This study provided a new RF raw material useful for improving and developing RF-based foods.

## 1. Introduction

Rice is one of the most widely grown and consumed cereal crops in the world [1]. In addition to being used as whole grains, rice, especially broken rice, is also commonly processed into rice flour (RF), which is frequently included in cakes, noodles, pasta, and other RF-based foods [2,3]. Starch and protein, which account for 80–90% and 5–10%, respectively, of the dry weight of RF, determine the processing method and quality of the final products. Rice protein has a reasonable amino acid composition, high digestibility, and nutritional value when compared with the protein from other major cereals, such as wheat and corn [4]. Furthermore, rice protein is shown to have excellent physiological functions, such as inflammation-reducing, anticancer, and antioxidant [5]. However, the protein content of RF is comparatively low, and to get enough protein, too much rice and easily digested starch are consumed, which may cause obesity and increase the risk of type II diabetes. Therefore, it is desirable to increase rice protein content in RF.

According to the origin of the protein, the reinforcement can be divided into either exogenous reconstitution or endogenous enrichment. At present, exogenous reconstitution is the main way, and protein concentrates/isolates, such as rice protein [6], whey proteins [7], wheat proteins [8], etc., are used to reconstitute RF or rice starch (RS). In the case of rice proteins, the concentrates or isolates are mainly prepared by the alkaline method, which negatively affects the quality of the reconstitution due to some drawbacks, such as dark color and toxic compounds such as lysinoalanine [4]. In endogenous enrichment, the starch is enzymatically degraded [9] or physically separated [10] to increase the relative content of protein in RF. Due to expensive devices and tedious procedures, physical enrichments, such as microfluidization [10], supercritical fluid extrusion [11], and ultrasound [12,13], are of limited value. For starch degradation, RF is commonly gelatinized at a high temperature and then hydrolyzed by thermostable α-amylase and amyloglucosidase [14,15]. However, because the starch granule structure is destroyed during the gelatinization, a series of physicochemical properties are lost, particularly pasting behaviors and endothermic characteristics, which reduces the application in food processing. Furthermore, the enriched protein is denatured [16], and side effects such as Maillard reactions [17] lower the nutritional value. In addition, genetic breeding and nitrogen-rich cultivation are commonly used ways to increase the protein content in RF. At present, a commercial high-protein rice variety named Frontière (average protein content is 11%), is available and has been used in the preparation of gluten-free muffins, bread, and cupcakes [18,19,20].

However, with starch-degrading enzymes developing, hydrolysis of raw starch at moderate temperatures could avoid the starch gelatinization step. The key enzymes in this heterogeneous hydrolysis are called granular starch hydrolyzing enzymes (GSHE) or raw starch degrading enzymes (RSDE), including α-amylase, β-amylase, and amyloglucosidase [21,22]. There are many advantages to using GSHE, such as low energy consumption, cost savings, and the absence of side reactions. At present, GSHE, especially from commercial sources, has mostly been explored and applied in the field of fuel ethanol [23] and enzymatically modified starch [22,24]. While numerous studies report on the effects of enzymatic modification on the structural and physicochemical properties of starch, detailed insight into the mechanisms of degradation for different enzymes is lagging. Nevertheless, the previous application of 4-α-glucanotransferase to multicomponent food raw materials such as RF [25] provides an example and inspiration for us to improve the endogenous rice protein content in RF.

In RF, conceivably, the presence of non-starch components (protein and fat) will restrict the efficiency of starch hydrolysis by the GSHE, the key factor determining the extent of elevation of the endogenous rice protein content. Therefore, it is important to understand the mechanism of GSHE to improve the efficiency of enzymatic hydrolysis. Rice starch (RS) is the major component in RF and the substrate of GSHE. In RF, RS displays a multi-scale structure mainly including six levels: individual branches (level 1), amylopectin and amylose (level 2), semi-crystalline lamellas (level 3), growth rings (level 4), starch granules (level 5), and whole-grain structure (level 6) [26]. Enzymatic hydrolysis alters the multi-scale structure of RS and simultaneously improves the relative protein content, eventually influencing the physicochemical properties of RF. In the food field, processability can be defined as the behavior and interaction between food components during the processing stages, such as transportation, mixing, stirring, extrusion, and various heat treatments, which can be reflected by physicochemical properties [27,28,29]. In order to improve protein content under the premise of maintaining the processing properties, GSHE was selected to modify RS and RF in this study. During the process, the structural and physicochemical properties of the RF and RS prepared from RF were determined to propose a hydrolytic mechanism and create a new RF raw material for improving traditional or developing new RF-based foods.

## 2. Results and Discussion

### 2.1. The Degree of Hydrolysis (DH), Total Starch, and Total Protein Contents 

The DH, total starch, and total protein content of native and partially hydrolyzed RF and RS (Table 1) showed that RS and RF were highly degraded after the first 1 h to DH of 21.02% and 24.15%, and that DH finally reached 33.95% and 44.54% for RF-24 and RS-24, respectively, after incubation for 24 h. For the RF samples, the total starch content decreased significantly, while the total protein content increased significantly by the GSHE hydrolysis. Compared with RF-N, the protein of RF-24 increased by 54.58% after hydrolysis for 24 h, indicating an effective enrichment of the endogenous rice protein. Additionally, the DH in the RF group was significantly lower than for the RS group at the same hydrolysis time except for 1 h. The phenomenon of reduced hydrolysis of RF compared with RS was also observed by in vitro digestion of the starch in RF and RS and was associated with the physical barrier between starch and the digestive enzyme formed by non-starch components, such as protein and fat [30,31]. The result indicated the feasibility of using GSHE to degrade starch granules and to improve the protein content of RF.

### 2.2. Morphological Structure and Particle Size Distribution 

The micromorphology and particle size distribution of native and partially hydrolyzed RF and RS are shown in Figure 1 and Figure 2, and the d_50_ (the median diameter), D_[4,3]_ (De Broncker mean diameter), D_[3,2]_ (Sauter mean diameter), and span factor are summarized in Table 2. For RF-N, starch granules with smooth surfaces and polyhedral shapes were closely arranged to form agglomerates with non-starch components such as protein and fat [32], resulting in a wide size range (0.40–110.00 μm) with a unimodal distribution, similar to other studies of RF made by dry milling [33]. After GSHE hydrolysis, numerous pinholes and some pores appeared on the granule surface as shown in RF-1 (Figure 1). Meanwhile, some tightly packed agglomerates were loosened, leading to the dispersion of some starch granules and changing the size distribution from unimodal to trimodal, showing two new peaks consistent with the peaks of the RS group (Figure 2). The span factor of the RF group increased with the increase in hydrolysis time, and all values were higher than those in the RS group, in which it showed no significant change, indicating that the dispersion of starch granules from the agglomerate surface reduced the uniformity of RF. With increasing hydrolysis time, the size of holes and pores was enlarged and d_50_, D_[4,3]_, and D_[3,2]_ decreased significantly. After 24 h of hydrolysis, the porous appearance was reduced or even disappeared from the surface of the starch granules (RF-24), but different sorts of pits and craters arose and changed the surface from smooth to rough. This trend of surface variation was similar for the RF and RS groups, but the change in the RS group was more dramatic. However, for the RS group, d_50_ and D_[3,2]_ only decreased slightly, while D_[4,3]_ and the span factor did not change significantly, indicating that the hydrolysis only occurred on the surface of the starch granules. In addition, a unique phenomenon of massive hydrolysis was observed in the RS group, occurring simultaneously on a single starch granule, leading to the formation of numerous pore structures or hollow shells (circled by the red lines in Figure 1), which might be related to the higher degree of hydrolysis in RS.

The action of the GSHE on the starch granules can be divided into three steps: diffusion, adsorption, and finally hydrolysis through a heterogeneous catalytic mechanism on the granular surfaces [34]. Warren et al. [35] believed that there are some disorganized regions, such as amorphous regions, on the surface of starch granules, in which disordered molecules are preferentially hydrolyzed leading to different morphological changes on the surface. According to the effect of enzymatic hydrolysis on the surface of starch granules, five erosion patterns are observed: pinholes, sponge-like erosion, numerous medium-sized holes, distinct loci leading to single holes in individual granules, and surface erosion [22]. In this study, we found that GSHE first resulted in pinhole erosion, and then spread around the surface of the granules, creating numerous medium-sized holes and eventually turning into surface erosion. During the whole process, the granule size decreased, and agglomerates in RF loosened or were even broken. The surface of starch granules contained a range of holes and pores to pit structures, indicating a change to an “outside-in” from an “inside-out” hydrolysis mode [36]. In conclusion, GSHE exhibited unique erosion patterns on the surface of starch granules and different effects on the agglomerates of RF depending on the hydrolysis times, which forms a basis for regulating the physicochemical properties of RF-based foods.

### 2.3. Crystalline Structure and Molecular Order

To further understand the erosion patterns of GSHE on the surface of starch granules, the crystalline structure and molecular order of the RF and RS groups were analyzed by wide-angle X-ray diffraction (XRD) and Fourier transform infrared spectroscopy (FTIR) (Figure 3). The relative crystallinity (RC), R_1047/1022_ (usually used to indicate the level of the short-range order of starch) [37], and R_995/1022_ (used to evaluate the double helix order in starch crystalline regions) [37] were calculated (Table 3). Both the RF and RS groups showed typical A-type XRD pattern (Figure 3A), characterized by strong diffraction peaks at 15°, 17°, 18°, and 23° [38] and a weak diffraction peak at 20° due to the V-amylose-lipid complex [30]. After the hydrolysis by GSHE, the RF, and RS showed almost no change in the crystallization patterns. The relative crystallinity of RF was higher than that of RS at the same hydrolysis time, but both slightly increased after GSHE treatment, indicating only a minor change in the crystalline structure of RF and RS with GSHE hydrolysis, which was consistent with the previous report [39]. Considering the DH, it can be suggested that hydrolysis occurred in the alternating crystalline and amorphous regions. This result was contrary to previous research, in which enzymes preferentially degraded the poorly organized amorphous areas, which relatively increased the RC [40,41].

Information on the molecular order obtained from the deconvoluted FTIR curves (Figure 3B) indicated consistency of the RF and RS groups, but the peak strength of the RF group was significantly lower than that of the RS group, probably due to the presence of non-starch components that mainly accumulate on the surface in RF samples [30]. Moreover, it was worth noting that R_1047/1022_ slightly increased in the RF group but decreased in the RS group with increasing hydrolysis time compared with the native counterparts, although no significant difference was observed except for RF-24. The reduction in RS-1, RS-6, and RS-24 in the RS group possibly stemmed from the erosion effect exerted by the GSHE, leading to more disordered starch granules’ surfaces. For the RF group, the dissociation of some starch granules on the agglomerate surface exposed the starch granules encased inside, which were not hydrolyzed by GSHE and showed a higher short-range order, while hydrolyzed starch granules demonstrated a decreased order. Therefore, the changes in molecular order for the RF group were more complex.

### 2.4. Amylose Content and Amylopectin Chain Length Distribution

Amylose and amylopectin constitute the fine molecular structure of starch that determines the basis of the multi-scale structure and influences the physicochemical properties of RF and RS [42]. Therefore, to investigate the enzyme degradation on amylose and amylopectin in detail, RS was hydrolyzed by GSHE for 1, 3, 6, 9, and 24 h; native RS, named RS-N, and only RS annealed for 24 h without the addition of GSHE, named RS-C, were used as the control. The amylose content and the chain length distribution of native, annealed, and partially hydrolyzed RS are shown in Table 4. The RS-N contained 17.99% amylose, which decreased to 13.98% (RS-C) after 24 h of annealing treatment due to amylose leaching [43]. The amylose content did not change significantly by hydrolysis of 1 and 3 h but decreased thereafter to 10.98% (RS-24) after hydrolysis of 24 h, probably related to amylose leaching through pores and holes after the enzyme GSHE hydrolysis and expanded annealing time. Leached amylose was available and easily degraded by GSHE [40].

Furthermore, while the chain length distribution was not affected only by annealing treatment for 24 h (Table 4), it distinctly varied with the hydrolysis time. Compared with RS-N and RS-C, the relative content of very short chains (DP < 6, DP represents the degree of polymerization) of RS-1 and RS-3 increased about 5 times, while the different groups of DP > 6 decreased to varying degrees (Table 4). Similar results were observed after the degradation of GSHE on pre-swelled maize starch mainly due to rapid hydrolysis of B chains (DP 13–36) [44]. At 6–24 hydrolysis, the stability of GSHE was reduced, and hydrolysis products accumulated, leading to gradually decreasing hydrolysis of the medium- and long-branch chains. Some very short chains were hydrolyzed during the end of the reaction, resulting in a relative increase in the content of the DP > 6 chains.

### 2.5. Thermal Properties

Thermal treatment is one of the most common RF-based food processing methods used for rice paste, rice bread, and rice noodles, and the thermal properties of native and partially hydrolyzed RF and RS during gelatinization were determined by differential scanning calorimetry (DSC) (Figure 4 and Table 5). Interestingly, all RF samples presented two endothermic peaks, while a single peak was found in the RS group (Figure 4), which was also observed, but not explained by Cappa et al. [45]. It may be related to the agglomerates of numerous closely packed starch granules in RF (Figure 1), which caused uneven internal and external heat absorption and melting. RF-N had higher T_o1_ and T_p1_ compared with RS-N, indicating that the complete gelatinization of RF required a higher temperature.

With longer hydrolysis time compared with RS-N, the T_o1,_ and T_p1_ of RS samples increased significantly, and T_c1_-T_o1_ decreased. The ΔH in DSC curves showed the amount of heat energy required to melt the crystalline region of starch, reflecting the ratio of crystalline to amorphous regions [46]. The ΔH_1_ of RS-1 and RS-6 slightly decreased but showed no significant difference compared with RS-N. However, RS-24 had the highest ΔH_1_, which was consistent with the highest relative crystallinity in the RS group (Table 3). However, as for the RF group, the effect of thermal properties presented was more complicated during the hydrolysis. Some starch granules dissociated from the surface of agglomerates after enzymatic treatment in RF samples (Figure 1 and Figure 2), so the first endothermic peak of the RF group, the single starch endothermic peak, shifted to the left, specifically exhibiting significantly lower To1 and T_p1_ of RF-1 than RF-N. The increase in ΔH_1_ of GSHE-treated RF could be elucidated by the dispersion of starch granules from the aggregate surface due to the higher ΔH_1_ of RS than RF. However, RF-24 could be an exception, where its ΔH_1_ was slightly higher than RF-N, but without significant difference. Conceivably, the second endothermic peak in the RF group hydrolyzed by GSHE was almost unaffected, represented by slightly reduced T_o2_, T_p2_, and T_c2_ (shown in Appendix A), which could also be explained by some starch granules escaping from the agglomerate surface. All in all, compared with starch gelatinization and degradation at a high temperature to improve rice protein content, the RF partially hydrolyzed by GSHE maintained appropriate thermal processing properties as a raw material to meet the current production requirements of many RF-based foods.

### 2.6. Pasting Behavior

Pasting is widely used in the thermal processing of various food materials containing starch, such as extrusion, baking, cooking, enzyme treatment, and re-crystallization and the viscosity variation of RF and RS during pasting can be described by rapid viscosity analysis (RVA) [47]. Compared with RS-N, RF-N had a lower viscosity curve (Figure 5) due to the protein and fat content, which was consistent with previous studies [48]. After hydrolysis by GSHE, peak viscosity (PV), through viscosity (TV), breakdown viscosity (BV), and final viscosity (FV) of the RF and RS groups (Table 6) showed a similar tendency. For example, the PV of RF-1 and RS-1 rapidly declined to 37.54% of RF-N and 33.91% of RS-N, respectively, and the PV of RF-24 and RS-24 partially recovered to 51.89% of RF-N and 65.84% of RS-N, respectively, indicating that GSHE hydrolysis could affect the viscosity of RF during pasting. The phenomenon was probably related to the changes in the starch molecular structure (Table 4) and the increase in the relative content of rice protein (Table 1). In the early stages of GSHE hydrolysis, the increase in very short chains in amylopectin reduced the cross-linked interactions between starch components [49]. In addition, rice protein in the RF group inhibited starch swelling, playing a significant role in the viscosity drop [50]. With the continuation of hydrolysis, the relative content of DP > 6 components, especially the DP > 36 components (Table 4) which were regarded as the backbone of amylopectin, increased, promoting cross-linking interactions between starch molecules [51]. In addition, the significant decrease in amylose content also has a positive effect on recovery in pasting viscosity [52]. The degree of recovery in pasting viscosity of RF-24 and RS-24 was significantly greater for the RS group compared with the RF group, which indicated that the increased protein content in RF also had a positive impact on the thermal processing of RF-based foods. Furthermore, the pasting temperature (PT) of RF increased compared with that of the native sample after hydrolysis by GSHE. However, no significant difference was determined in RF samples treated with GSHE at different times. These results showed that a higher temperature was required to gelatinize RF-based foods made by GSHE-treated RF.

Setback viscosity (SV) could reflect the degree of short-term retrogradation of paste during cooling after thermal processing due to molecular rearrangement [53]. The SV in the RS samples had a similar tendency as the other viscosity parameters, but RS-6 and RS-24 gave higher SV values than RS-N after 6 h, while RF-1, RF-6, and RF-24 had approximately the same, albeit lower SV values than RF-N (Table 6). This difference indicated that the short-term recovery of starch gels was promoted by prolonged GSHE hydrolysis but delayed by the increase in protein content in rice paste. Rice protein could improve the water-holding capacity and hinder water migration during the retrogradation of rice starch gels [54].

### 2.7. Frequency Sweep

The frequency sweep of pastes formed by native and partially hydrolyzed RF and RS (Figure 6) can reflect the gel characteristics of the pastes that are important for their practical production and application [55]. The storage modulus (G’), loss modulus (G’’), and loss factor (Tan (θ)) with angular frequency in both RF and RS groups had similar variation tendencies but different variation scales. For all samples, G’ was always greater than G’’, and Tan(θ) was between 0 and 0.5, all increasing with the angular frequency increasing, which indicated that the pastes after RVA tests displayed weak gels with a slight frequency dependency [56]. Compared with RF-N and RS-N, the G’ and G’’ of the RF and RS group, hydrolyzed for 1 h by GSHE, decreased sharply but increased to a varying degree after 6 h and 24 h, similar to the variation trends of the RVA curves (Figure 5). Especially for RS-24, G’ was about 4 times as high as for RS-N, and G’’ was slightly higher than for RS-N, suggesting that the elastic component increased substantially and the viscous component was restored to the original level. However, all samples of hydrolyzed RF had similar G’ and G’’, but all were lower than for RF-N, confirming that the viscosity and elasticity of hydrolyzed RF were reduced. Furthermore, the Tan(θ) (<1) of both RF and RS groups decreased with longer hydrolysis time, which displayed a tendency similar to RF baked at higher temperatures due to the degradation of starch molecules during the baking [56]. In our study, the GSHE effectively hydrolyzed starch and changed the amylose content and branch chain distribution of amylopectin. At the early stage of hydrolysis, the relative content of very short chains (DP < 6) was remarkably augmented, which weakened the interactions with other starch molecules or branch chains. At the later stage of hydrolysis, the amylose content decreased, and the content of middle- and long-chain components (DP > 24), as the backbone in amylopectin, increased. Generally speaking, the amylose content had a positive correlation with G’ or G’’ and a negative correlation with Tan(θ) [52]. However, in this study, the erosion by GSHE mainly occurred on the surface of starch granules, while amylose, especially more weakly bound to amylopectin molecules in the starch granules, leached more with erosion from holes to pits, resulting in the residual amylose in the granules being able to interact more strongly with other molecules, such as the backbone and branches of amylopectin.

### 2.8. Discussion

In this study, RF and isolated RS were hydrolyzed at 55 °C by GSHE (containing α-amylase and amyloglucosidase) aiming to improve the protein content in RF by partially removing starch. During hydrolysis, GSHE exhibited unique erosion patterns on starch granules, which significantly changed the granular and molecular structure and affected starch agglomerates in RF, which was related to the mode of action of α-amylase and amyloglucosidase.

The relative hydrolytic activity of α-amylase and amyloglucosidase during hydrolysis of RS by GSHE was compared by measuring the composition and relative content of oligosaccharides released, based on the fact that exo-acting amyloglucosidase only releases glucose. The results in Appendix A combined with results on hydrolysis degree (Table 1) and chain length distribution (Table 4) indicated that with the increase in hydrolysis time, the total hydrolytic activity of GSHE was continuously decreased, but the relative activity of endo-acting α-amylase decreased faster than that of the exo-acting amyloglucosidase. As shown in Figure 7, for RF-1, due to the rapid hydrolysis by α-amylase and amyloglucosidase, the surface of starch granules was porous, and the crystallized and amorphous areas were hydrolyzed simultaneously, which decreased the peak viscosity and gelatinization temperature range. However, the accumulation of very short chains (DP < 6) increased significantly, which weakened the interaction between molecules, resulting in a remarkable decrease in the viscoelastic properties of rice paste. However, for RF-24, the hydrolysis rate was weakened, and pits formed on the surface of starch particles due to the erosion of GSHE. Meanwhile, the slow hydrolysis of α-1,6 glucosidic bonds by amyloglucosidase resulted in the increased relative content of DP >24 chains, which interacted strongly with other molecules, such as residual amylose, leading to the recovery of pasting viscosity and viscoelastic properties. However, the degree of change in RF was obviously weaker than that of RS due to the improvement of protein content and lower hydrolysis degree. Compared with RF-N, these changes in physicochemical properties, especially for RF-24, indicated that the processability of RF with high protein content was partially maintained, and it could be used as a raw material for the potential applications of various RF-based foods, such as rice bread and noodles.

## 3. Materials and Methods

### 3.1. Materials

Dry-milled Indica RF, made in Taizhou, Jiangsu Province (purchased online) was passed through a 100-mesh sieve for later use. The experimental enzyme mixture (Novozyme 5009) was generously provided by Novozymes Co. Ltd. (Copenhagen, Denmark), including fungal α-amylase (EC 3.2.1.1, GH 13) and amyloglucosidase (EC 3.2.1.3, GH 15) and named GSHE in this work. The activity (8111 U/mL) was assayed at 55 °C with 1.0% (*w*/*v*) RS suspension as substrate in 50 mM sodium acetate, pH 5.0, using the 3,5-dinitrosalicylic acid (DNS) method to quantify released reducing sugars [57] and glucose as standard. One U was defined as the amount of enzyme forming 1.0 mg of reducing sugar per mL under the above conditions. Total starch assay kit (K-TSTA), amylose/amylopectin assay kit (K-AMYL), and isoamylase (E-ISAMY, 200 U/mL) were purchased from Megazyme Co. Ltd. (Wicklow, Ireland). All other reagents, solvents, and chemicals were reagent-grade and obtained from Sinopharm Chemical Reagent Co., Ltd. (Shanghai, China).

### 3.2. RS Isolation

RS was isolated following the method of Liu et al. [58] with slight modification. RF was mixed with 0.2% (*w*/*v*) NaOH 1:4 (*w*/*v*), stirred continuously at room temperature for 4 h, and centrifuged (5000 rpm, 10 min). The yellow protein layer on top of the precipitate was discarded, and the residue was washed 3 times using 0.2% (*w*/*v*) NaOH until the yellow layer disappeared. The slurry was neutralized to pH 7 by 1 M HCl and washed 3 times with deionized water. The final starch sediment was dried at 45 °C for 48 h, ground in a mortar, and passed through a 100-mesh sieve.

### 3.3. Preparation of Hydrolyzed RF and RS by GSHE

RS (20.0 g, dry basis) and RF (25.5 g, dry basis, equivalent to 20.0 g RS dry basis) were suspended in 100 mL 50 mM sodium acetate, pH 5.0, and preheated in a water bath shaker (55 °C, 30 min). GSHE (50 U/g dry starch) was added to the RF and RS slurries and incubated for 1, 6, and 24 h (55 °C, continuous shaking at 160 rpm). The reaction was terminated by placing the slurries in an ice bath to cool instantly and rapidly adding 3 mL 1 M NaOH. After 15 min with intermittent shaking, pH was adjusted back to 5.0 by adding 3 mL 1 M HCl, and the slurry was centrifugated (5000 rpm, 10 min). The supernatant was collected to analyze the degree of hydrolysis (DH), and the sediment was washed 3 times using abundant deionized water. Finally, the hydrolyzed RS and RF were dried (45 °C, 48 h), ground in a mortar, and passed through 100-mesh sieves. The samples were named RS-1, RS-6, RS-24, RF-1, RF-6, and RF-24. Native samples without any treatment were named RS-N and RF-N.

Furthermore, to investigate the hydrolytic mechanism of GSHE in detail, RS was also hydrolyzed by GSHE under the above process for 3 and 9 h, and the obtained samples were named RS-3 and RS-9. RS subjected to the above process was used as the control, named RS-C, but without the addition of GSHE (only continuously shaken at 55 °C for 24 h as annealing treatment).

### 3.4. Determination of Total Starch, Total Protein, Amylose Contents, DH, and the Released Oligosaccharides

The total starch and amylose content were determined using a total starch assay kit (K-TSTA) and an amylose/amylopectin assay kit (K-AMYL), respectively. The total protein content was determined by an automatic Kjeldahl nitrogen analyzer (K1160, Hanon Advanced Technology Group Co., Ltd., Jinan, China) using the factor 5.95 to convert nitrogen content to crude protein content.

DH was determined by the amount of reducing sugar in the supernatant measured by the DNS method [57] using the equation below. Briefly, the supernatant was suitably diluted by deionized water, and 0.5 mL was mixed with 0.5 mL DNS reagent, boiled for 5 min, and transferred to an ice-water bath to rapidly cool down to R.T. The absorbance was measured at 540 nm using a microplate reader (SPECTRA MAX 190, Molecular Devices, San Jose, CA, USA). The standard curve was made by replacing the diluted supernatant with the standard glucose solution (0.0–0.5 mg/mL).
(1)$$\mathrm{DH}\;\left(\%\right)=\frac{\mathrm{Tr}\times 0.9}{\mathrm{Ms}}\times 100$$
where Tr is the weight of reducing sugar in the supernatant expressed as glucose, 0.9 is the glucose-to-starch conversion factor, and Ms is the weight of dry starch.

The relative composition and content of released oligosaccharides were determined by high-performance liquid chromatography (HPLC, Waters e2695, Water, Milford, MA, USA) equipped with X-Bridge BEH Amide column (250 mm × 4.6 mm). The supernatant of hydrolyzed starch was diluted with ultrapure water, and the same volume of pure acetonitrile was added and centrifugated (10000× *g*, 30 min). Acetonitrile solution (65%, *w*/*w*) was used as the mobile phase with 0.8 mL/min flow rate, the column temperature was 30 °C, and the loading volume was 50 μL. Glucose (G1), maltose (G2), maltotriose (G3), maltotetraose (G4), maltospentaose (G5), maltohexaose (G6), and maltoheptaose (G7) were used as the standard.

### 3.5. Particle Size Distribution

Particle size distribution of all samples was determined using the laser particle size analyzer (BT-9300 ST, Dandong Baxter Instrument Co., Ltd., Dandong, China). Deionized water was used as a dispersant, and the sample flour was added gradually to the sample pool. The sample was completely dispersed by ultrasound 3 times for 3 s each time. The measurement was carried out under the conditions of a pump speed of 1600 rpm and a shielding range of 10–15%. The refractive indexes of water and samples were 1.52 and 1.33, respectively [59]. The measurement was carried out in triplicate, and d_50_, D_[4,3]_, D_[3,2]_, and the span factor were recorded.
(2)$$\mathrm{Span}\;\mathrm{factor}\;=\frac{d_{90}-d_{10}}{d_{50}}$$
where the d_10_, d_50_, and d_90_ mean of the cumulative particle diameters represent 10%, 50%, and 90% of the entire range of size.

### 3.6. Scanning Electron Microscopy (SEM)

The micromorphology was captured by scanning electron microscope (SEM, SU8100, Hitachi High-Tech, Tokyo, Japan). Flour adhered to the conductive tape, and gold was sprayed onto the flour in a vacuum. The final observation was conducted at ×3000 magnification operating at a low accelerating voltage of 15 kV and irradiation voltage of 1 kV.

### 3.7. XRD

XRD was conducted by using a diffractometer (D2 PHASER, Bruker AXS, Karlsruhe, Germany) equipped with Cu-Kα radiation and operated with scanning from 4–40° (4.79 °/min) at 0.04° with a count time of 0.5 s. RC was calculated by using Jade 6.0 software (Material Date, Inc., Livermore, CA, USA) as previously described [60].

### 3.8. FTIR

FTIR spectra were obtained by an FTIR spectrometer (IS10, Thermo Nicolet Inc., Waltham, MA, USA) equipped with an ATR accessory in the region of 400–4000 cm^−1^ with a resolution of 4 cm^−1^ by 64 scans. The air background spectrum was deducted from each tested sample. The region of 1200−800 cm^−1^ was chosen to analyze the short-range order of samples with OMNIC 8.0, the half-bandwidth, and enhancement factor were set as 19 cm^−1^ and 1.9, respectively. The ratios between intensities at 1047 and 1022 cm^−1^ (R1047/1022) and 995 and 1022 cm^−1^ (R995/1015) over deconvoluted spectra were calculated. 

### 3.9. HPAEC

The branch chain length distribution of RS samples was determined by high-performance anion exchange chromatography with pulsed amperometric detection (HPAEC-PAD, ICS-5000+, Thermo Fisher Scientific, Waltham, MA, USA). Starch (10 mg) was suspended in 5 mL 50 mM sodium acetate, pH 4.5, and boiled for 30 min to be completely gelatinized. The gelatinized starch was fully debranched by 2 U isoamylase (40 °C, 12 h). After inactivation by boiling for 10 min and centrifugation (10,000× *g*, 10 min), the supernatant was filtered (0.22 μm membrane filter), and 20 μL was injected onto a CarboPac PA200 column at 30 °C, eluted at a flow rate of 0.4 mL/min using isocratic 150 mM NaOH and a linear gradient of 0–400 mM sodium acetate as mobile phase [49].

### 3.10. DSC

The thermal property of the samples was measured by a calorimeter (DSC7000, HITACHI, Japan). First, 3 mg of sample powder and 6 mg of deionized water were added to an aluminum pan. After a slight shaking, sample pans were sealed and equilibrated overnight at 4 °C. The temperature was linearly increased from 30 to 100 °C° at 10 °C/min. An empty aluminum pan was used as a reference. To, Tp, Tc, and ΔH were reported for each endothermic peak.

### 3.11. RVA

The pasting properties of samples were determined using a rapid viscosity analyzer (RVA-Starch Master2, Perten Instruments, Stockholm, Sweden). Deionized water was added to each sample (3.0 g on a 14% moisture basis) to a total weight of 28 g, and the analysis was conducted as previously described [60]. Viscosity parameters, including PV, TV, BV, FV, and SV, were obtained through the built-in software.

### 3.12. Rheological Measurement

The rheological characterization was done using a rheometer (Discovery hybrid rheometer, TA Instruments, New Castle, DE, USA) with a parallel-plate system (40 mm diameter) and a working gap of 1000 mm. After the gelatinization of samples by RVA (described in Section 3.11), the paste was rapidly cooled to 25 °C for 10 min. The paste was subjected to a frequency sweep test from 0.1 to 100 rad/s at a target strain of 1% (in the linear viscoelastic region) at 25 °C, and G’, G’’, and Tan (θ) were determined.

### 3.13. Statistic Analysis

Experimental results are reported as the average of triplicate measurements. The statistical significance was assessed by Tukey’s test using OriginPro 2022 b (OriginLab, Northampton, MA, USA). A *p*-value <0.05 was considered to be statistically significant throughout the study.

## 4. Conclusions

By controlling the reaction time of GSHE to RF, significant endogenous enrichment of the rice protein was achieved, while appropriate processing properties of RF were maintained. After the 24 h hydrolysis, the relative rice protein content improved from 8.52% to 13.17%. The results of the microstructure, crystal structure, and molecular structure of the RF indicated that the GSHE treatment rapidly increased the very short chains of amylopectin through the early-stage pinhole erosion, which then changed to pit erosion, further hydrolyzing the very short chains and increasing the relative content of the middle and long chains. In the whole process, the crystalline and amorphous regions were hydrolyzed, causing a slight increase in relative crystallinity. This hydrolytic mechanism led to a sharp decrease in viscosity during pasting after 1 h of hydrolysis followed by a slow increase, similar to the change in the viscoelastic properties of the pastes, but without significantly changing the thermal properties, which confirms that the physicochemical properties of RF can be properly preserved. The study brings more insights into structure–function relationships during the GSHE hydrolysis and creates a new RF raw material with high protein content by endogenous enrichment for application in various RF-based food, such as rice noodles and reconstituted rice, especially for gluten-free food, such as muffins, cakes, and bread.

## Figures and Tables

**Figure 1 molecules-28-03522-f001:**
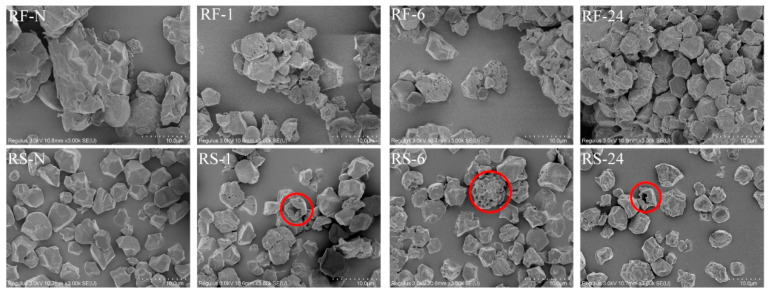
Morphological structure of native and partially hydrolyzed RF and RS at ×3000 magnification. The upper left abbreviations are sample names, in which RF represents rice flour and RS represents rice starch; numbers refer to the hydrolysis time (h); RF-N is native rice flour and RS-N is native starch isolated from rice flour. The red lines circle the numerous pore structures or hollow shells formed by simultaneous hydrolysis on a single starch granule.

**Figure 2 molecules-28-03522-f002:**
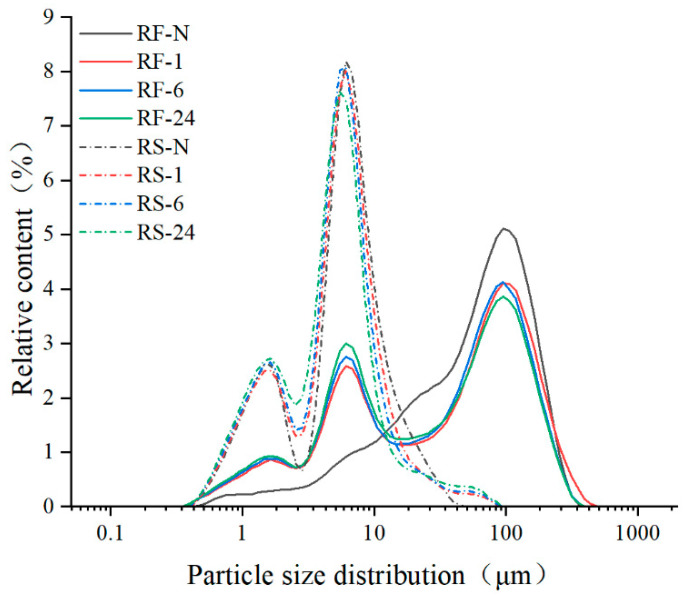
The particle size distribution of native and partially hydrolyzed RF and RS. RF represents rice flour and RS represents rice starch; numbers refer to the hydrolysis time (h); RF-N is native rice flour and RS-N is native starch isolated from rice flour.

**Figure 3 molecules-28-03522-f003:**
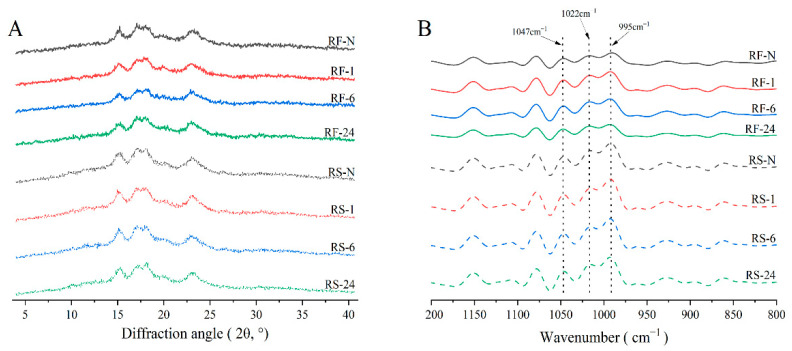
The X-ray diffraction (**A**) and deconvoluted FTIR (**B**) of native and partially hydrolyzed RF and RS. RF represents rice flour and RS represents rice starch; numbers refer to the hydrolysis time (h); RF-N is the rice flour and RS-N is native starch isolated from rice flour.

**Figure 4 molecules-28-03522-f004:**
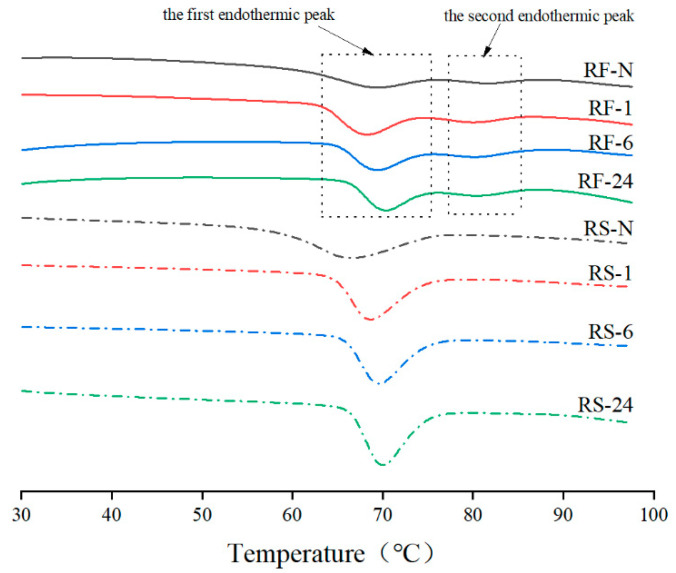
Differential scanning calorimetry (DSC) curves of native and partially hydrolyzed RF and RS. RF represents rice flour and RS represents rice starch; numbers refer to the hydrolysis time (hours); RF-N is native rice flour and RS-N is native starch isolated from rice flour. The two endothermic peaks of the RF group were pointed out.

**Figure 5 molecules-28-03522-f005:**
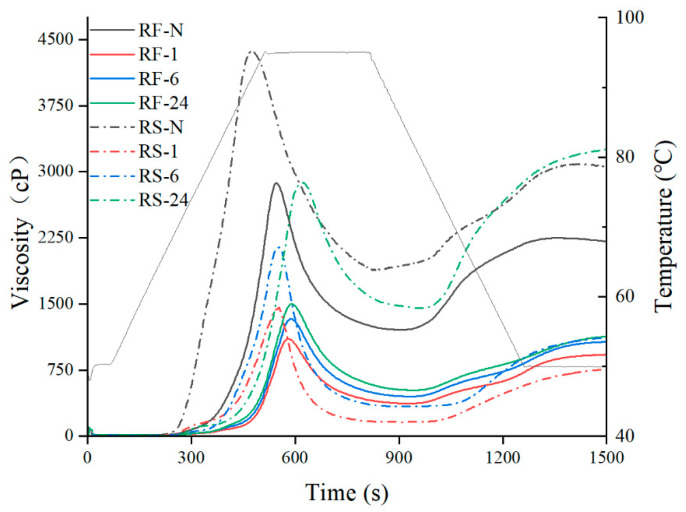
The RVA curves of native and partially hydrolyzed RF and RS. RF represents rice flour and RS represents rice starch; numbers refer to the hydrolysis time (h); RF-N is native rice flour and RS-N is native starch isolated from rice flour.

**Figure 6 molecules-28-03522-f006:**
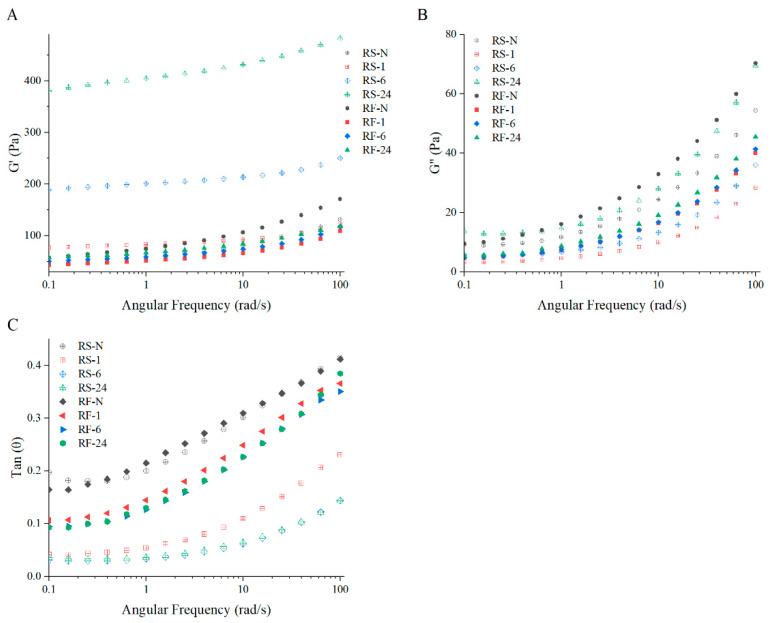
Viscoelastic characteristics of pastes formed by native and partially hydrolyzed RF and RS: variation of G’ (**A**) and G’’ (**B**) and Tan(θ) (**C**) with angular frequency for samples. G’, storage modulus; G’’, loss modulus, Tan (θ), loss factor. RF represents rice flour and RS represents rice starch; numbers refer to the hydrolysis time (hours); RF-N is native rice flour and RS-N is native starch isolated from rice flour.

**Figure 7 molecules-28-03522-f007:**
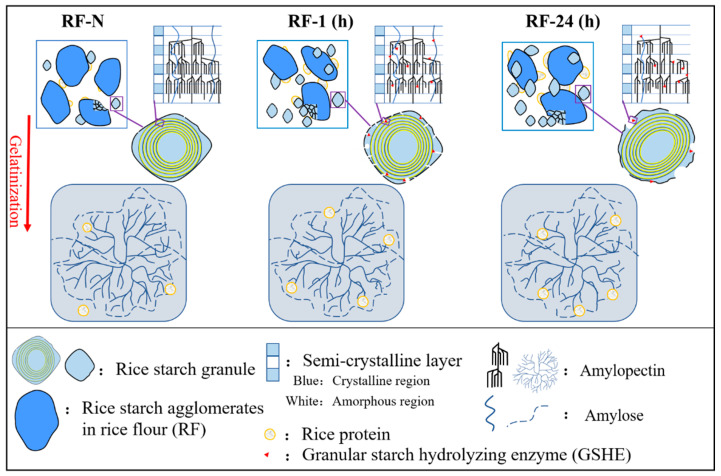
The diagram of the hydrolytic mechanism of GSHE on RF.

**Table 1 molecules-28-03522-t001:** The degree of hydrolysis (DH) and total starch and total protein contents ^1^ of native and partially RF and RS.

Samples ^2^		DH (%)	Total Protein (%)	Total Starch (%)
RF	RF-N	—	8.52 ± 1.12 ^c^	89.90 ± 0.06 ^a^
	RF-1	24.15 ± 0.15 ^c^	10.57 ± 0.31 ^b^	86.37 ± 0.27 ^b^
	RF-6	29.41 ± 0.51 ^b^	11..20 ± 0.19 ^b^	84.05 ± 0.16 ^c^
	RF-24	33.95 ± 0.12 ^a^	13.17 ± 0.35 ^a^	82.22 ± 0.62 ^d^
RS	RS-N	—	0.34 ± 0.08	98.27 ± 0.88
	RS-1	21.02 ± 0.68 ^C^	—	—
	RS-6	33.43 ± 0.31 ^B^	—	—
	RS-24	44.54 ± 0.53 ^A^	—	—

^1^ Means ± SD values followed by different lowercase letters of the RF group or different uppercase letters of the RS group are significantly different (*p* < 0.05). “—” means no test. ^2^ RF represents rice flour and RS represents rice starch; numbers refer to the hydrolysis time (hours); RF-N is native rice flour and RS-N is native starch isolated from rice flour.

**Table 2 molecules-28-03522-t002:** The particle size parameters ^1^ of native and partially hydrolyzed RF and RS.

Samples ^2^		d_50_ (μm)	D_[4,3]_ (μm)	D_[3,2]_ (μm)	Span Factor
RF	RF-N	63.26 ± 0.16 ^a^	75.47 ± 0.13 ^a^	15.18 ± 0.04 ^a^	2.46 ± 0.01 ^d^
	RF-1	45.90 ± 0.45 ^b^	66.22 ± 0.66 ^b^	7.34 ± 0.02 ^b^	3.49 ± 0.01 ^c^
	RF-6	41.11 ± 0.78 ^c^	60.77 ± 1.82 ^c^	7.04 ± 0.04 ^c^	3.62 ± 0.04 ^b^
	RF-24	33.14 ± 0.22 ^d^	56.90 ± 0.02 ^d^	6.56 ± 0.00 ^d^	4.34 ± 0.04 ^a^
RS	RS-N	5.99 ± 0.12 ^A^	7.50 ± 0.84 ^A^	3.29 ± 0.10 ^A^	2.25 ± 0.26 ^A^
	RS-1	5.50 ± 0.02 ^B^	7.02 ± 0.10 ^A^	3.10 ± 0.01 ^AB^	2.05 ± 0.06 ^A^
	RS-6	5.20 ± 0.06 ^C^	6.77 ± 0.05 ^A^	2.95 ± 0.04 ^B^	1.97 ± 0.06 ^A^
	RS-24	4.88 ± 0.00 ^D^	6.76 ± 0.04 ^A^	2.83 ± 0.00 ^C^	2.08 ± 0.01 ^A^

^1^ Means ± SD values followed by different lowercase letters of the RF group or different uppercase letters of the RS group are significantly different (*p* < 0.05). ^2^ RF represents rice flour and RS represents rice starch; numbers refer to the hydrolysis time (h); RF-N is native rice flour and RS-N is native starch isolated from rice flour.

**Table 3 molecules-28-03522-t003:** The XRD and FTIR parameters ^1^ of native and partially hydrolyzed RF and RS.

Samples ^2^		RC (%)	R_1047/1022_	R_995/1022_
RF	RF-N	19.72 ± 0.91 ^a^	0.5373 ± 0.0539 ^a^	1.4243 ± 0.0599 ^a^
	RF-1	19.83 ± 0.83 ^a^	0.5862 ± 0.0158 ^a^	1.2673 ± 0.0711 ^ab^
	RF-6	21.03 ± 0.49 ^a^	0.5868 ± 0.0153 ^a^	1.2871 ± 0.0864 ^ab^
	RF-24	21.19 ± 0.18 ^a^	0.6599 ± 0.0095 ^b^	1.2145 ± 0.0313 ^b^
RS	RS-N	18.44 ± 0.28 ^A^	0.5841 ± 0.0178 ^A^	1.5158 ± 0.0814 ^A^
	RS-1	18.62 ± 0.26 ^A^	0.5773 ± 0.0026 ^A^	1.4989 ± 0.0739 ^A^
	RS-6	19.32 ± 0.86 ^A^	0.5639 ± 0.0055 ^A^	1.5079 ± 0.0255 ^A^
	RS-24	20.21 ± 0.39 ^A^	0.5605 ± 0.0155 ^A^	1.4506 ± 0.0888 ^A^

^1^ Means ± SD values followed by different lowercase letters of the RF group or different uppercase letters of the RS group are significantly different (*p* < 0.05). RC, R_1047/1022_, and R_995/1022_ represent the relative crystallinity, the short-range order of starch, and the double helix order in starch crystalline regions, respectively. ^2^ RF represents rice flour and RS represents rice starch; numbers refer to the hydrolysis time (h); RF-N is native rice flour and RS-N is native starch isolated from rice flour.

**Table 4 molecules-28-03522-t004:** The amylose content ^1^ and chain length distribution of amylopectin in native, control, and partially hydrolyzed RS.

Samples ^2^	Amylose Content (%)	Chain Length Distribution (%)				
		DP < 6	DP 6–12	DP 13–24	DP 25–36	DP ≥ 36
RS-N	17.99 ± 0.28 ^A^	3.65	31.29	53.16	10.20	1.69
RS-C	13.98 ± 0.41 ^C^	3.69	31.03	53.11	10.52	1.65
RS-1	17.79 ± 0.13 ^AB^	18.30	28.71	43.01	8.57	1.41
RS-3	17.86 ± 0.91 ^A^	19.87	27.79	42.55	8.41	1.39
RS-6	17.37 ± 0.25 ^AB^	18.89	28.45	42.72	8.55	1.38
RS-9	16.37 ± 0.08 ^B^	15.60	28.56	45.43	8.81	1.60
RS-24	10.98 ± 0.40 ^D^	13.91	28.30	47.11	9.04	1.65

^1^ Means ± SD values followed by different uppercase letters of the RS group are significantly different (*p* < 0.05). ^2^ RS represents rice starch; numbers refer to the hydrolysis time (h); RS-N is native starch isolated from rice flour; and RS-C is the control subjected to annealing treatment for 24 h.

**Table 5 molecules-28-03522-t005:** The DSC parameters ^1^ of native and partially hydrolyzed RF (the first endothermic peak) and RS.

Samples ^2^		T_o1_ (°C)	T_p1_ (°C)	T_c1_ (°C)	T_c1_-T_o1_ (°C)	ΔH_1_ (mJ/mg)
RF	RF-N	63.51 ± 0.01 ^c^	69.47 ± 0.18 ^b^	72.33 ± 0.15 ^b^	8.81 ± 0.17 ^a^	4.40 ± 0.57 ^b^
	RF-1	65.62 ± 0.69 ^b^	68.31 ± 0.08 ^c^	71.13 ± 0.18 ^c^	5.51 ± 0.55 ^b^	6.42 ± 0.41 ^a^
	RF-6	66.45 ± 0.18 ^b^	69.14 ± 0.28 ^b^	71.95 ± 0.05 ^b^	5.51 ± 0.14 ^b^	6.52 ± 0.14 ^a^
	RF-24	67.99 ± 0.20 ^a^	70.15 ± 0.19 ^a^	73.39 ± 0.35 ^a^	5.40 ± 0.15 ^b^	5.31 ± 0.11 ^b^
RS	RS-N	59.61 ± 0.21 ^C^	66.64 ± 0.29 ^B^	72.95 ± 0.14 ^AB^	13.34 ± 0.24 ^A^	13.35 ± 0.32 ^B^
	RS-1	66.11 ± 0.02 ^B^	68.59 ± 0.03 ^A^	72.50 ± 0.08 ^B^	6.39 ± 0.10 ^B^	12.60 ± 0.26 ^B^
	RS-6	67.34 ± 0.02 ^A^	69.60 ± 0.06 ^A^	73.07 ± 0.20 ^A^	5.73 ± 0.19 ^BC^	12.77 ± 0.37 ^B^
	RS-24	67.73 ± 0.27 ^A^	69.57 ± 0.78 ^A^	73.20 ± 0.29 ^A^	5.47 ± 0.55 ^C^	14.34 ± 0.23 ^A^

^1^ Means ± SD values followed by different lowercase letters of the RF group or different uppercase letters of the RS group are significantly different (*p* < 0.05). T_o_: onset temperature; T_p_: peak temperature; T_c_: conclusion temperature; ΔH: enthalpy change. Number “1” represents the first endothermic peak in the RF group; ^2^ RF represents rice flour and RS represents rice starch; numbers refer to the hydrolysis time (h); RF-N is the native rice flour control and RS-N is the native starch control isolated from rice flour.

**Table 6 molecules-28-03522-t006:** The RVA parameters ^1^ of native and partially hydrolyzed RF and RS.

Samples ^2^		PV (cP)	TV (cP)	BV (cP)	FV (cP)	SV (cP)	PT (°C)
RF	RF-N	2868.67 ± 9.45 ^a^	1223.00 ± 13.45 ^a^	1645.67 ± 16.80 ^a^	2230.00 ± 17.35 ^a^	1007.00 ± 4.00 ^a^	76.52 ± 0.81 ^b^
	RF-1	1077.00 ± 27.51 ^d^	345.33 ± 30.55 ^d^	731.67 ± 7.09 ^d^	894.33 ± 28.57 ^d^	549.00 ± 7.94 ^d^	82.68 ± 0.23 ^a^
	RF-6	1316.00 ± 38.74 ^c^	431.33 ± 26.50 ^c^	884.67 ± 15.82 ^c^	1061.67 ± 28.92 ^c^	630.33 ± 9.71 ^b^	82.33 ± 0.96 ^a^
	RF-24	1488.67 ± 22.28 ^b^	525.67 ± 8.14 ^b^	963.00 ± 19.97 ^b^	1131.00 ± 15.10 ^b^	605.33 ± 10.79 ^c^	81.12 ± 0.90 ^a^
RS	RS-N	4383.33 ± 9.50 ^A^	1888.00 ± 14.73 ^A^	2495.33 ± 18.04 ^A^	3061.33 ± 9.61 ^B^	1173.33 ± 9.07 ^B^	69.88 ± 0.03 ^B^
	RS-1	1486.67 ± 25.54 ^D^	164.67 ± 5.69 ^D^	1322.00 ± 20.88 ^D^	779.00 ± 14.73 ^D^	614.33 ± 11.59 ^C^	73.32 ± 0.36 ^A^
	RS-6	2165.67 ± 20.64 ^C^	304.50 ± 21.92 ^C^	1579.00 ± 32.53 ^B^	1502.00 ± 107.48 ^C^	1197.50 ± 85.56 ^B^	73.52 ± 0.80 ^A^
	RS-24	2886.00 ± 4.24 ^B^	1437.00 ± 22.63 ^B^	1449.00 ± 26.87 ^C^	3240.00 ± 18.38 ^A^	1803.00 ± 4.24 ^A^	74.68 ± 0.86 ^A^

^1^ Means ± SD values followed by different lowercase letters of the RF group or different uppercase letters of the RS group are significantly different (*p* < 0.05). PV: peak viscosity; TV: through viscosity; BV: breakdown viscosity; FV: final viscosity; SV: setback viscosity; PT: pasting temperature. ^2^ RF represents rice flour and RS represents rice starch; numbers refer to the hydrolysis time (h); RF-N is native rice flour and RS-N is native starch isolated from rice flour.

## Data Availability

Data are contained within the article or Appendix A.

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
