# Peer review of "Increasing Protein Content of Rice Flour with Maintained Processability by Using Granular Starch Hydrolyzing Enzyme"

_molecules, 2023, doi:10.3390/molecules28083522_

Round 1
Reviewer 1 Report
The manuscript describes the enhancement of the endogenous protein content in rice fluor (RF) by using a granular starch hydrolyzing enzyme (GSHE) to create a new RF with comparable processing properties. Additionally, the authors propose a hydrolytic mechanism resulting from the action of GSHE. The manuscript could be considered for publication after addressing the following comments.
Major comments:
Please provide additional information regarding the potential application of the resulting RF after GSHE treatment.
Please discuss a relationship between hydrolytic mechanism of GSHE and the mode of action of -amylase and amyloglucosidase.
Please elaborate the correlation between the span factor and the coefficient of variation of the particle size distribution observed both in RF and RS.
Please discuss the increase in entalphy observed in GSHE-treated RF (except for RF-24), while GSHE-treated RS showed a decrease in entalphy (except for RS-24) compared to the control groups.
It would be beneficial if the authors could create a diagram to illustrate structure-function relationship during the GSHE hydrolysis.
Minor comments;
No red circular was found in Figure 1.
Please provide additional details on how particle size distribution was determined in the “Material and Methods” section.
Please rewrite the following sentence in order to improve its clarity. “….indicating only a minor change in the crystalline structure, reminiscent of a former report, in which starch in RF …”
Reviewer 2 Report
The manuscript “Increasing protein content of rice flour with maintained processability of starch by using granular starch hydrolyzing enzyme (Molecules-2275342)” investigated the properties of rice flour prepared by a granular starch hydrolyzing enzyme. In general, this research might be able to provide a method to provide a new rice flour material. However, there are observations which need to be justified in the manuscript.
1. In Table 2, 3, 5, and 6, the characteristics of RF and RS should be compared (RF-N and RS-N, RF-1 and RS-1, ……) to give more information.
2. Please give the specific discussion on the â–³H1 in table 5.
3. Please give the definition of processability of starch. Does the processability mean the physicochemical properties?
4. Why is processability of starch in title, while processing properties of RF in the conclusion?
5. The results of RVA curves and viscoelastic property changed significantly by the enzyme. Is it maintained?
Round 2
Reviewer 2 Report
1. The following response to Q2 should be improved. “The ΔH1 for RS-1 and RS-6 slightly decreased, while there was no significant difference for RS-1 and RS-6.”
2. What are the gelatinization temperatures determined by RVA?
3. The following response to Q5 should be improved. “Unlike cooked or partially gelatinized RF, for example, the use of extruded flour requires the addition of a larger volume of water to obtain a constant consistency during dough-making due to partial gelatinization [1].” It is contradictory for the two partial gelatinization statements.
Round 3
Reviewer 2 Report
Fine.